# Excess Risk of Major Adverse Cardiovascular and Kidney Events after Acute Kidney Injury following Living Donor Liver Transplantation

**DOI:** 10.3390/jcm11113100

**Published:** 2022-05-30

**Authors:** Yi-Chia Chan, Cheng-Hsi Yeh, Lung-Chih Li, Chao-Long Chen, Chih-Chi Wang, Chih-Chi Lin, Aldwin D. Ong, Ting-Yu Chiou, Chee-Chien Yong

**Affiliations:** 1Liver Transplantation Center, Department of Surgery, Kaohsiung Chang Gung Memorial Hospital, Kaohsiung 833, Taiwan; littlewhalesparkle@gmail.com (Y.-C.C.); ycc9002108@gmail.com (C.-H.Y.); clchen@cgmh.org.tw (C.-L.C.); ufel4996@gmail.com (C.-C.W.); immunologylin@gmail.com (C.-C.L.); aldwin.ong@gmail.com (A.D.O.); 2Liver Transplantation Center, Department of Internal Medicine, Division of Nephrology, Kaohsiung Chang Gung Memorial Hospital, Kaohsiung 833, Taiwan; longee01@gmail.com (L.-C.L.); cyang@cgmh.org.tw (T.-Y.C.)

**Keywords:** acute kidney injury, liver transplantation, major adverse cardiovascular events, major adverse kidney event, cardiovascular disease

## Abstract

Acute kidney injury (AKI) is a well-known risk factor for major adverse kidney events (MAKE) and major adverse cardiovascular events (MACE) in nontransplant settings. However, the association between AKI after liver transplantation (LT) and MACE/MAKE is not established. A retrospective cohort analysis including 512 LT recipients was conducted. The incidence of post-LT AKI was 35.0% (*n* = 179). In total, 13 patients (2.5%) developed de novo coronary artery disease (CAD), 3 patients (0.6%) diagnosed with heart failure (HF), and 11 patients (2.1%) had stroke. The post-LT AKI group showed a higher incidence of CAD and HF than the no post-LT AKI group (4.5% versus 1.5%, *p* = 0.042; 1.7% versus 0%, *p* = 0.018; respectively), while there was no significant difference in the stroke events (2.8% versus 1.8%, *p* = 0.461). Through Cox regression analysis, history of cardiovascular disease (HR 6.51, 95% CI 2.43–17.46), post-LT AKI (HR 3.06, 95% CI 1.39–6.75), and pre-LT diabetes (HR 2.37, 95% CI 1.09–5.17) were identified as independent predictors of MACE; pre-LT chronic kidney disease (HR 9.54, 95% CI 3.49–26.10), pre-LT diabetes (HR 3.51, 95% CI 1.25–9.86), and post-LT AKI (HR 6.76, 95% CI 2.19–20.91) were risk factors for end-stage renal disease. Post-LT AKI is predictive for the development of MACE and MAKE.

## 1. Introduction

With the rapid progress and development in the field of liver transplantation (LT), short-term and long-term graft outcomes have continued to improve; nonetheless, five-year mortality rates amongst LT recipients remain relatively high at 25 to 30% [1]. The main causes of patient mortality include infections and recurrent disease; these commonly present early after LT. Conversely, malignancy, cardiovascular disease (CVD), and renal failure typically develop gradually over time [2,3], highlighting the importance of long-term postoperative care. In addition, there is growing evidence indicating that major adverse cardiovascular events (MACE) and major adverse kidney events (MAKE) not only increase the healthcare burden with higher rates of re-hospitalization and healthcare utilization [4,5], but also pose a negative impact on patient survival [3,6].

Several risk factors have been recognized to be associated with the development of MACE and MAKE irrespective of surgical procedure [7,8,9]. In nontransplant populations, postoperative acute kidney injury (AKI) has been observed to be a risk factor for both MACE and MAKE [7,8,9]. Large epidemiological cohort studies reported that AKI survivors are at increased risk of kidney and cardiovascular dysfunction in the short-term and long-term follow-up [7,10,11]. In addition, the incidence of post-LT AKI remained high and might exceed 50% in some series [12,13], depending on the definitions used. Even patients who merely experience a brief period of postoperative AKI have thus far shown poor recipient and graft outcomes [14].

Although renal complications in LT survivors with AKI are well recognized, such as prolonged hospital length of stay and higher rates of CKD and end-stage renal disease (ESRD), it remains unclear whether AKI confers an additional detrimental risk of MACE after living donor LT (LDLT) [9,15]. This retrospective study was therefore performed with the aim of examining the long-term cardiovascular and renal outcomes in recipients who sustain AKI after liver transplantation.

## 2. Materials and Methods

### 2.1. Population

We retrospectively reviewed all patients who underwent LDLT between January 2009 and December 2013 in Kaohsiung Chang Gung Memorial Hospital, Taiwan. Patients who were older than 20 years were included, and 527 patients were enrolled. Patients who were referred to other hospitals overseas after transplantation (*n* = 7) or lost to follow-up (*n* = 4) and those who died within the first 72 h after LT (*n* = 4), were excluded, thus leaving 512 patients for the final analysis. The demographics, diagnoses, and pre- and perioperative parameters of the recipients are described in Table 1.

### 2.2. Definition

The Kidney Disease Improving Global Outcomes (KDIGO) classified AKI into three stages according to the increase of serum creatinine (Scr) [16]: stage 1 referred to an increase in Scr ≥ 0.3 mg/dL within 48 h or 1.5–1.9 times baseline within 7 days, stage 2 referred to an increase in Scr of 2.0–2.9 times baseline within 7 days, stage 3 referred to an increase in Scr of >3.0 times baseline or ≥4.0 mg/dL, with an acute increase of at least 0.5 mg/dL, or the need for renal replacement therapy within 7 days. Urine output was not recorded in this study and was not included in the classification of AKI. Post-LT AKI is based on changes in serum creatinine from baseline creatinine within the first week after transplantation [17]. Chronic kidney disease (CKD) was defined as having an estimated glomerular filtration rate (eGFR) < 60 mL/min per 1.73 m^2^ using the Chronic Kidney Disease Epidemiology Collaboration (CKD-EPI) equation for three consecutive months [18]. Coronary artery disease (CAD) was defined as any degree of stenosis in any of the coronary arteries, determined by visual inspection using coronary angiography [19]. Heart failure (HF) was defined as a patient with symptoms of cardiac failure and at least one echocardiographic abnormality [20]. MACE included cardiovascular death, myocardial infarction, coronary revascularization, heart failure, new atrial fibrillation, cardiac arrest, or pulmonary embolism [21]. In this study, we used CAD to include myocardial infarction and coronary revascularization. MAKE was defined as a composite of CKD and ESRD [22]. Additionally, patients with pre-LT CKD were categorized as MAKE when their eGFR dropped more than 25% from baseline [22].

### 2.3. Study Information

The demographics of the recipients, along with preoperative, operative, and postoperative variables, were collected. Demographic information included the following: age and sex of the recipients, underlying etiology of liver disease, and diagnosis of hepatocellular carcinoma (HCC). Preoperative clinical variables included the following: past medical history of diabetes mellitus (DM), hypertension (HTN), CKD, history of CVD including CAD or stroke, Model of End-stage Liver Disease (MELD) score, recipient body mass index (BMI), liver function, Scr level, and the estimated glomerular filtration rate (eGFR) before LT. Perioperative data collected included the following: blood loss during transplantation and estimated graft recipient weight ratio (GRWR). Postoperative data included daily Scr measurements up to one week after LT, which was used to define post-LT AKI.

The primary outcome of this study was the probability of developing MACE and MAKE in patients who had post-LT AKI. The secondary outcome was the all-cause mortality rate after LDLT with increasing stages of post-LT AKI. This study was approved by the Institutional Review Board (No. 201801436B0). The need for informed consent from patients was waived because of the retrospective design of the study.

### 2.4. Preoperative Cardiovascular Risk Stratification

As part of our evaluation protocol, EKG and echocardiography procedures were conducted before LDLT. A thallium scan was conducted for LT candidates over 40-years-old, with an existing history of CAD, or abnormal cardiac exam. Specialized cardiologists assessed the perioperative cardiovascular risk for every patient. In moderate- to high-risk patients, percutaneous coronary intervention (PCI) was done. As per protocol, transplant surgery was postponed for at least four weeks in patients who underwent PCI stenting followed by dual antiplatelet therapy. Patients diagnosed with HF or intractable CAD (multiple small coronary arteries, precluding intervention) were advised against undergoing LT. Recipients with a history of stroke were assessed with brain MRI and carotid duplex scan and referred to a specialized neurologist for stroke risk stratification.

### 2.5. Immunosuppression, Postoperative Care, and Surveillance for Post-LT AKI

The standard immunosuppression protocol after LDLT involved intravenous (IV) basiliximab (Simulect; Novartis Pharma AG, Basel, Switzerland) on post-LT Days 1 and 4. Steroid therapy consisted of intraoperative methylprednisolone (500 mg/IV) bolus dose, titrated to 20 mg/day, which was then switched to prednisolone 20 mg/tablet/day once the oral intake was tolerated. The steroid dose was gradually tapered and discontinued over 3 months if no acute cellular rejection developed. Tacrolimus combined with mycophenolate mofetil (MMF) and corticosteroids was given to all patients as initial immunosuppression therapy. The oral dose of tacrolimus was adjusted daily to achieve a target trough level of 8–10 ng/mL during the first two weeks following surgery, and then 5–8 ng/mL thereafter. Some patients with advanced HCC, or patients who were noted to develop post-LT AKI were converted to a mammalian target of rapamycin (mTOR) inhibitor.

After LT, all recipients stayed in the intensive care unit for two weeks to closely monitor volume status and renal function. To prevent AKI, albumin replacement, fluid infusion therapy, and diuretics were administered. They were followed up every one to three months after discharge until the end of the study date (31 December 2021) or the date of death. At the outpatient clinic, patients were closely monitored for their hepatic and renal function, and any symptoms or signs of developing MAKE and MACE. The mean follow-up duration was 9.3 ± 2.9 years.

### 2.6. Statistical Analysis

Numerical data are expressed as the mean (standard deviation) and were compared using independent *t*-tests. Categorical data are expressed as rates or proportions and were compared using the chi-square test. Cox regression analysis was used to evaluate the relationship between AKI and the time to occurrence of MACE, ESRD, and death. Variables with *p* values less than 0.2 were considered significant in the univariate analysis and were included in the multivariate analysis. The estimated probability of MACE or ESRD events was analyzed and graphed using the Kaplan–Meier method, and groups were compared using the log-rank test. Results were considered significant if the *p*-value was <0.05. Statistical analysis was performed using SPSS software v.20 (SPSS, Chicago, IL, USA).

## 3. Results

### 3.1. Population Characteristics

The study population included 512 LDLT recipients. A total of 179 patients (35.0%) developed post-LT AKI after LDLT, while no AKI was evident in 333 patients (65.0%). Among patients who had post-LT AKI, 135 patients (75.4%) were classified as stage 1 AKI; 36 patients (20.1%), as stage 2 AKI; and 8 patients (4.5%), as stage 3 AKI. The demographic, clinical, and pre- and peri-operative variables, as well as the results of a comparison between both study groups, are shown in Table 1.

All recipients were initiated to tacrolimus immediately after LDLT, and as expected, a significantly larger proportion of recipients who developed post-LT AKI were also subsequently started on mTOR inhibitors compared to patients without post-LT AKI (81.6% vs. 61.0%, *p* < 0.001). There were 26 patients (5.1%) with CKD and 19 patients (3.7%) with CVD in this cohort before transplantation (Table 1). However, there was no significant difference in the occurrence of post-LT AKI in patients with a history of CKD or CVD.

### 3.2. Major Adverse Cardiac Events

During the mean follow-up of 9.3 ± 2.9 years, 26 (5.3%) patients were diagnosed with MACE, including de novo CAD, HF, and stroke, as listed in Table 2. No case of new-onset atrial fibrillation, pulmonary embolism, or unknown cardiac arrest was documented. Compared with the no post-LT AKI group, the AKI group had a higher rate of de novo CAD (4.5% vs. 1.5%; *p* = 0.02) and HF (1.7% vs. 0%; *p* = 0.018). However, there was no difference in the incidence of stroke between the two groups (2.8% vs. 1.8%; *p* = 0.461). Three patients had new-onset HF, and all expired because of severe cardiac dysfunction. Classifying the etiology of HF, one case was ischemia type due to triple vessel CAD, and another two cases were mitral valve regurgitation related.

The likelihood of patients developing MACE after post-LT AKI was 2.3% at year 2, 6.2% at year 5, and 9.0% at year 10 after transplantation. The corresponding likelihoods for patients without AKI were 0.9%, 1.2% and 3.2%, respectively, by using the Kaplan–Meier method (Figure 1). In univariable analysis, older age (hazard ratio [HR] 1.06, 95% CI 1.00–1.13, *p* = 0.047), history of CVD (HR 6.85, 95% CI 2.58–18.18, *p* < 0.001), post-LT AKI (HR 3.07, 95% CI 1.41–6.72, *p* = 0.005), pre-LT DM (HR 2.62, 95% CI 1.23–5.78, *p* = 0.013), post-LT CKD (HR 3.74, 95% CI 1.67–8.39, *p* = 0.001) and ESRD (HR 6.08, 95% CI 2.09–17.67, *p* = 0.001) were associated with the development of MACE. In the multivariable analysis, however, history of CVD (HR 6.51, 95% CI 2.43–17.46, *p* < 0.001), post-LT AKI (HR 3.06, 95% CI 1.39–6.75, *p* = 0.006), and pre-LT DM (HR 2.37, 95% CI 1.07–5.17, *p* = 0.030) were significant predictors of MACE (Table 3).

### 3.3. Major Adverse Kidney Events

As shown in Table 2, the overall prevalence of CKD was 28.3% (145/512) in our cohort. The incidence of CKD was higher in the post-LT AKI group compared with no post-LT AKI group, either at three months after LT or at the end of follow-up (22.3% vs. 6.3%, *p* < 0.001; 44.1% vs. 19.8%, *p* < 0.001; respectively).

Eighteen (3.5%) patients were newly diagnosed with ESRD after the liver transplant surgery and needed dialysis therapy or renal transplantation. The likelihood of developing ESRD after post-LT AKI was 2.3% at year 2, 3.7% at year 5, and 10.0% at year 10 after LT. The corresponding likelihoods for patients without post-LT AKI were 0.0%, 0.8%, and 1.3%, respectively (Figure 2A). Considering patients with both pre-LT CKD and post-LT AKI, the likelihood of progression to ESRD was 25% at 2 years, 35% at 5 years, and 44% at 10 years after LT. The corresponding likelihood were 0, 0, and 16.7% for those with pre-LT CKD only; 0.6%, 1.4%, and 7.0% for those with post-LT AKI only; and 0, 0.3%, and 0.7% for those with neither pre-LT CKD nor post-LT AKI (Figure 2B).

In univariable analysis, pre-LT diabetes (HR 5.51, 95% CI 2.04–14.91, *p* = 0.001), pre-LT CKD (HR 15.29, 95% CI 5.91–39.54, *p* < 0.001), and post-LT AKI (HR 8.01, 95% CI 2.63–24.43, *p* < 0.001) were associated with the occurrence of ESRD. In the multivariable analysis, pre-LT diabetes (HR 3.51, 95% CI 1.25–9.86, *p* = 0.017), pre-LT CKD (HR 9.54, 95% CI 3.49–26.10, *p* < 0.001), and AKI (HR 6.76, 95% CI 2.19–20.91, *p* = 0.001) were risk factors for ESRD (Table 4). Moreover, when post-LT AKI and pre-LT CKD were considered individually, patients with post-LT AKI were 10.17 times more likely to develop ESRD than those without injury, after adjustment for age, sex, DM, HTN, and CKD. However, when AKI and CKD occurred concurrently, the recipients were 76.36 times more likely to develop ESRD than those with neither condition.

### 3.4. All-Cause Mortality and Death Risk after Post-LT AKI

As shown in Table 2, the presence of post-LT AKI was associated with significantly increased mortality (25.1% versus 12.0%, *p* < 0.001). Using the KDIGO-equivalent AKI definition, we also observed that all-cause mortality increased with higher stages of severity, as shown in Figure 3. The mortality rate was 22.2% in AKI stage I, 33.3% in AKI stage 2, 37.5% in stage 3, and 50% in patients requiring renal replacement therapy (*p* < 0.001).

### 3.5. Risk Factors for Post-LT AKI

Recipients who developed post-LT AKI were noted to have been generally older, had higher MELD scores, less frequently presented with HCC, had pre-LT HTN, relatively lower preoperative serum albumin level, and had more blood loss than those in the no post-LT AKI group (*p* < 0.05) in Table 1. Using multivariate logistic regression analysis for risk factors for post-LT AKI, older age (odds ratio [OR] 1.04, 95% CI 1.01–1.06, *p* = 0.007), the presence of HCC (OR 0.64, 95% CI 0.43–0.93, *p* = 0.021), pre-LT HTN (OR 1.73, 95% CI 1.00–3.01, *p* = 0.045), blood loss (OR 1.05, 95% CI 1.02–1.08, *p* = 0.001), and pre-LT serum albumin level (OR 0.72, 95% CI 0.54–0.99, *p* = 0.041) were independent risk factors for AKI, as shown in Table 5.

## 4. Discussion

With improved LT surgical techniques and survival, the long-term care of LT recipients has become increasingly important [1]. Our study revealed a correlation of major adverse cardiovascular and kidney events with postoperative AKI among LT recipients with a median follow-up of 10 years. History of CVD, pre-LT DM, and post-LT AKI were predictors of future cardiac events, whereas pre-LT CKD, pre-LT DM, and post-LT AKI were risk factors for ESRD. Furthermore, underlying CKD patients complicated with post-LT AKI demonstrate a multiplicative risk for the development of ESRD.

AKI after liver transplantation is common, and the reported incidence ranges from 14% to 97% in deceased donor LT (DDLT) [13,14]. The variation in the post-LT AKI incidence rate comes from the different populations studied, different postoperative periods, and different definitions of AKI used. However, few studies have addressed AKI after LDLT, and Kwon et al. reported an incidence of 68% in a cohort of 1136 LDLT cases [12]. Another study by Hilmi et al. reported that LDLT recipients have a lower risk of post-LT AKI than DDLT recipients (23% versus 44.2%, respectively) within the first 72 h after LT, probably because of higher quality liver grafts [23]. In our cohort, the incidence of AKI during the first week after LDLT was 35.0%, and the result was comparable with previous studies.

Post-LT AKI is recognized as an independent risk factor for future MACE in nontransplant populations [8,10,11]. The reported frequency of long-term post-LT cardiovascular events varies greatly between different LT protocols and follow-up times. Our 10-years incidence of MACE is 5.3%, slightly lower than previous studies, reporting a range from 4.5% to 28% within three years after LT [24,25]. By disease-specific events, patients with post-LT AKI were associated with a higher incidence of CAD and HF than patients without AKI but not with the occurrence of stroke. Although some studies reported CKD was associated with the development of MACE, it was only evident in univariate analysis in our cohort. One of the explanations was the interaction between AKI and CKD. In multivariate analysis, underlying CVD, pre-LT DM, and post-LT AKI were found to be predictors of MAKE. Diabetes and pre-LT CAD are established determinates of MACE after LT [26,27], which correlated with our results. However, little information has addressed de novo cardiovascular disease in LT recipients with AKI. Although Faouzi et al. [25] proved that deteriorated renal function after transplantation increases the risk of cardiac events in a two-year prospective trial, AKI was not mentioned. Our findings proved the association between post-LT AKI and long-term cardiovascular events. Finally, all three heart failure patients expired, which also highlighted the importance of postoperative cardiovascular surveillance [28].

ESRD is a serious complication with a cumulative incidence between 1% and 4% within 5 years after LT [29,30], and is associated with a higher risk of mortality [31]. Trinh et al. [30] demonstrated that the incidence of new-onset ESRD after transplantation was significantly higher among patients who developed AKI. In our study, the cumulative probability of developing ESRD at 5 years was 3.7% and 0.8% for patients with and without post-LT AKI, also consistent with previous reports [29,30]. In addition to AKI, two other risk factors found in this study, underlying diabetes and CKD, were demonstrated to be as contributing determinants of future ESRD [32]. Moreover, recipients with both post-LT AKI and baseline CKD were far more likely to develop ESRD, suggesting a strong multiplicative effect of the interaction between the two conditions for the development of ESRD. Patients with impaired renal function have an increased risk for AKI [33], but the combined effect of AKI and CKD on long-term ESRD development has rarely been studied. In a prospective study of patients receiving dialysis, Metcalfe et al. [34] found that 44% required dialysis because of CKD, 36% because of AKI, and 20% because of acute on chronic renal failure, indicating that AKI superimposed on CKD induces a negative impact on the renal outcome.

Both pre-LT and de novo diabetes increase the risk of infection, CV events, ESRD, and death [35]. In our results, not only AKI, but also pre-LT DM was an independent predictor of future MACE and ESRD. Similarly, the associations between AKI and DM with the risk of renal failure were also reported in a large analysis including 69,321 transplants of nonrenal organs [36]. Pre-LT diabetes exhibited a more significant impact on mortality than de novo diabetes, which may imply the pre-existing condition influence the long-term outcome [2]. As in our results, pre-LT diabetes rather than de novo DM plays a crucial role in the CV and renal events.

Irrespective of the cause, the mortality rate is directly correlated with the stage and severity of AKI [37]. Although using different AKI definitions based on several criteria, the studies in which stages of AKI could be ascertained have demonstrated that patients with AKI experience a progressive increase in mortality with worsening AKI stages [37]. We used the KDIGO definition of AKI to classify our patients into three stages and subgroups needing renal replacement therapy (Figure 3). This result echoes those of other studies and proves that the mortality rate corresponds to the severity of AKI.

Several retrospective studies have demonstrated a wide variety of risk factors for the development of post-LT AKI, and its etiology is considered multifactorial origins encompassing recipients, grafts, and peri- and postoperative factors [13]. In this study, we found that aging, presence of HCC, HTN, hypoalbuminemia, and blood loss were correlated with post-LDLT AKI. The last two factors are consistent with the findings of Kwon et al. [12], as the number of units of transfused RBCs is regarded as a surrogate marker of blood loss. The presence of HCC is inversely associated with AKI occurrence, which may be explained by the lower Child score and reduced blood loss in these HCC patients [13]. Aging and HTN, although not reported in the LT setting, are well-known predictors for AKI in other non-LT hospitalized populations [38]. Moreover, whether immunosuppression therapy plays a role in the development of AKI cannot be proven in this study because CNI was used universally in every case and an mTOR inhibitor was prescribed when AKI occurred or in case of advanced HCC. Therefore, the use of mTOR inhibitors was regarded as a consequence of AKI events.

Summarizing the post-LT AKI risk factors in our study, nonmodifiable factors included patient characteristics (age, HTN) and hepatic status (hypoalbuminemia, HCC existence). Intraoperative blood loss, therefore, is probably the only modifiable factor. Hence, we should avoid major blood loss during LT to prevent long-term cardiovascular and renal complications. In addition, identifying these high-risk patients, as well as meticulous surveillance of CV and kidney function is recommended. Finally, early recognition of AKI by using biomarkers, avoidance of nephrotoxic drugs, and optimization of volume status and hemodynamics are also evident to prevent AKI-associated complications [39].

The study has several limitations. First, this was a retrospective observational analysis, which may impact the identification of some confounding factors. Second, we may underestimate the incidence of AKI because only serum creatinine values were used as the main criterion in the definition of AKI and urinary output was eliminated from the equation. Third, because this was a single-center study, our institutional perioperative strategy may influence the occurrence of post-LT AKI. Finally, the overall sample size was small and the median length of follow-up was only ten years. More studies with larger groups and longer follow-up periods are needed to further clarify the long-term consequences after post-LT AKI.

## 5. Conclusions

Compared with other studies discussing post-LT AKI, we provided a longer period of follow-up and showed a significant association between AKI and MACE/MAKE, as well as death. The mortality rate is directly correlated with the severity of AKI. Patients with both AKI and concurrent CKD showed a multiplicative risk for the development of ESRD. Prevention of AKI occurrence for high-risk patients is warranted to reduce the negative impact and to improve long-term recipient outcomes.

## Figures and Tables

**Figure 1 jcm-11-03100-f001:**
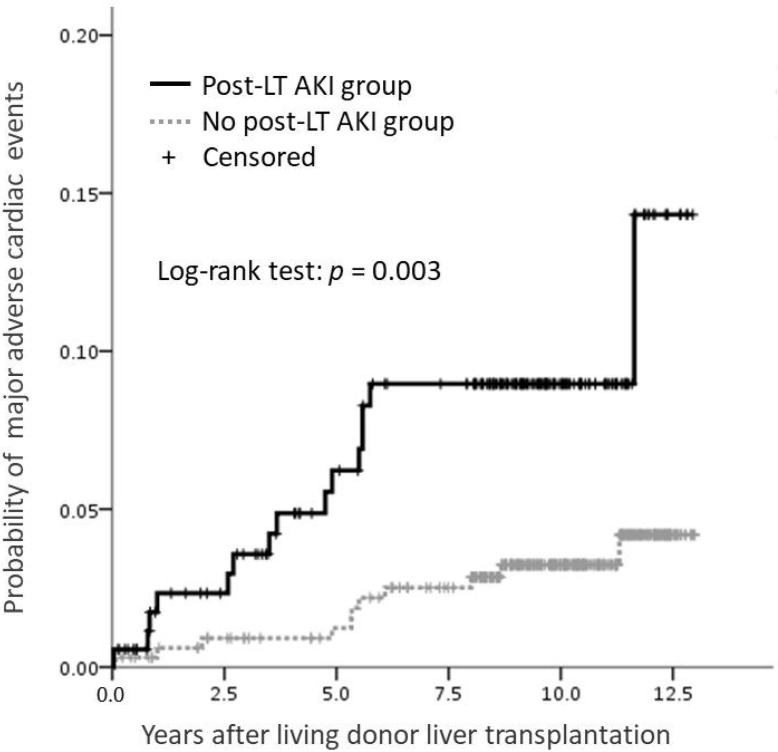
Estimated probability of major adverse cardiovascular events after living donor liver transplantation using the Kaplan–Meier method.

**Figure 2 jcm-11-03100-f002:**
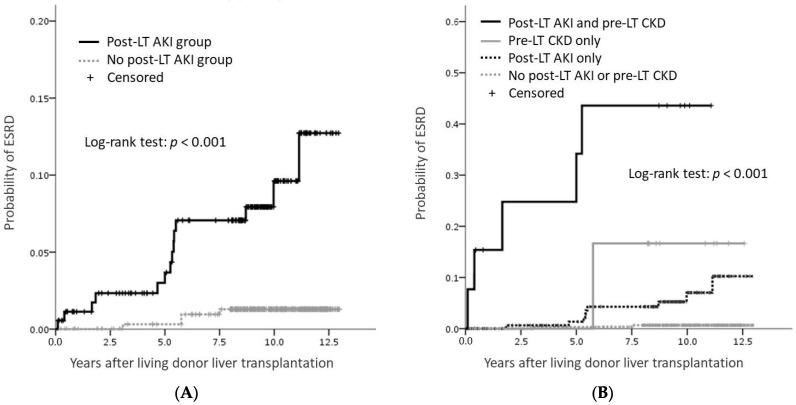
Estimated probability of ESRD occurrence after living donor liver transplantation using the Kaplan–Meier method. (**A**) Curves by post-LT AKI status. (**B**) Curves by post-LT AKI and pre-LT CKD status.

**Figure 3 jcm-11-03100-f003:**
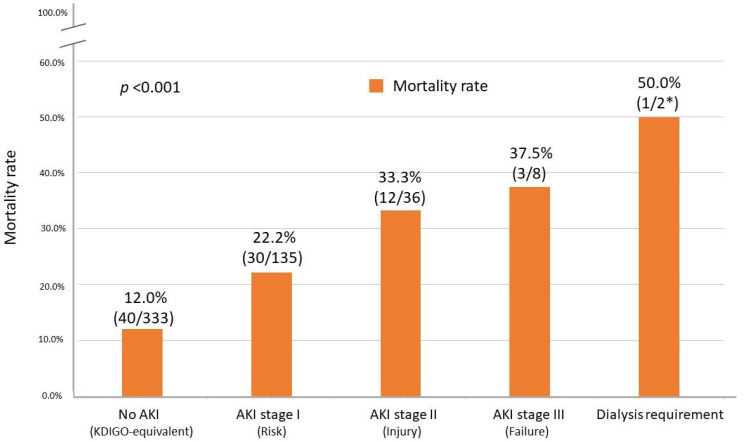
All-cause mortality rate after living donor liver transplantation with increasing stages of postoperative acute kidney injury. KDIGO—Kidney Disease Improving Global Outcome. * These post-LT AKI patients needing dialysis therapy were also included in the group of AKI stage III.

**Table 1 jcm-11-03100-t001:** Characteristics of the Patients Undergoing Living Donor Liver Transplantation.

	Total (*n* = 512)	Post-LT AKI (*n* = 179, 35.0%)	No Post-LT AKI (*n* = 333, 65.0%)	*p*-Value
Recipient age (years)	54.2 ± 8.1	55.4 ± 7.9	53.6 ± 8.1	0.018
Male sex, *n* (%)	378 (73.8%)	131 (73.2%)	247 (74.2%)	0.808
BMI (kg/m^2^)	24.9 ± 4.3	24.5 ± 4.6	25.1 ± 4.2	0.175
MELD	13.6 ± 7.6	14.5 ± 8.2	13.1 ± 7.3	0.041
Acute kidney injury				
Stage 0		-	333 (100%)	
Stage 1		135 (75.4%)	-	
Stage 2		36 (20.1%)	-	
Stage 3		8 (4.5%)	-	
Primary liver disease, *n* (%)				
Hepatitis B virus	240 (26.9%)	78 (43.6%)	162 (48.6%)	0.273
Hepatitis C virus	174 (34.1%)	60 (33.7%)	114 (34.2%)	0.521
Alcohol abuse	39 (7.6%)	17 (9.5%)	22 (6.6%)	0.240
Others	56 (10.9%)	23 (12.8%)	33 (9.9%)	0.310
HCC positive	277 (54.1%)	86 (48%)	191 (57.4%)	0.044
Preoperative comorbidities, *n* (%)				
Diabetes mellitus	120 (23.4%)	50 (27.9%)	70 (21.0%)	0.078
Hypertension	62 (12.1%)	29 (16.2%)	33 (9.9%)	0.037
HRS type I/II	5 (1.0%)/14 (2.7%)	3 (1.7%)/7 (1.9%)	2 (0.6%)/7 (2.1%)	0.238
Chronic kidney disease	26 (5.1%)	13 (7.3%)	13 (3.9%)	0.099
Cardiovascular disease	19 (3.7%)	9 (5.0%)	10 (3.0%)	0.248
Preoperative laboratory variables				
Serum albumin (g/dL)	3.1 ± 0.6	3.0 ± 0.5	3.1 ± 0.7	0.007
Serum total bilirubin (mg/dL)	1.3 ± 4.1	1.4 ± 3.2	1.2 ± 4.3	0.593
Serum creatinine (mg/dL)	0.8 ± 0.4	0.8 ± 0.4	0.8 ± 0.3	0.658
eGFR (mL/min/1.73 m^2^)	96.0 ± 24.6	95.4 ± 28.0	96.4 ± 22.6	0.679
Perioperative variables				
Blood loss (mL)	4443 ± 7228	6070 ± 9706	3577 ± 5179	0.002
GRWR	1.0 ± 0.2	0.9 ± 0.2	1.0 ± 0.2	0.449
Tacrolimus at initial CNI	511 (100%)	179 (100%)	332 (100%)	-
mTOR inhibitor use	349 (68.4%)	146 (81.6%)	203 (61.0%)	<0.001
Acute rejection, *n* (%)	143 (27.9%)	41 (22.9%)	102 (30.6%)	0.063

Values are expressed as means ± SD or numbers (percent). AKI—acute kidney injury; BMI—body mass index; CNI—calcineurin inhibitor; eGFR—estimated glomerular filtration rate; GRWR—graft to recipient weight ratio; HCC—hepatocellular carcinoma; HRS—hepatorenal syndrome; MELD—model of end-stage liver disease; mTOR—mammalian target of rapamycin.

**Table 2 jcm-11-03100-t002:** Long-term Cardiovascular and Renal Outcomes after Post-LT AKI.

Outcome	Total (*n* = 512)	Post-LT AKI (*n* = 179, 35.0%)	No Post-LT AKI (*n* = 333, 65.0%)	*p*-Value
Major adverse cardiac events (MACE)	26 (5.3%)	15 (8.4%)	11 (3.3%)	0.013
Coronary events	13 (2.5%)	8 (4.5%)	5 (1.5%)	0.042
Heart failure	3 * (0.6%)	3 * (1.7%)	0 (0.0%)	0.018
Stroke events	11 (2.1%)	5 (2.8%)	6 (1.8%)	0.461
Major adverse kidney events (MAKE)				
3-month CKD	61 (11.9%)	40 (22.3%)	21 (6.3%)	<0.001
Overall CKD	145 (28.3%)	79 (44.1%)	66 (19.8%)	<0.001
ESRD	18 (3.5%)	14 (7.8%)	4 (1.2%)	<0.001
Mortality	85 (16.6%)	45 (25.1%)	40 (12.0%)	<0.001
Follow up (years)	9.3 ± 2.9	8. 3 ± 3.3	9.8 ± 2.5	<0.001

CKD—chronic kidney disease; ESRD—end-stage renal disease. * One patient was diagnosed with coronary artery disease associated ischemia heart failure.

**Table 3 jcm-11-03100-t003:** Risk Factors of Major Adverse Cardiovascular Events after Living Donor Liver Transplantation by Cox Regression Analysis.

	Univariate Analysis			Multivariate Analysis		
	HR	95% CI	*p*-Value	HR	95% CI	*p*-Value
Age (year)	1.06	1.00–1.13	0.047			
Men	1.17	0.47–2.91	0.738			
BMI (kg/m^2^)	0.96	0.90–1.03	0.231			
Triglyceride (mg/dL)						
Before LT	1.00	0.99–1.01	0.631			
Post-LT 1st year	1.00	1.00–1.00	0.833			
Cholesterol (mg/dL)						
Before LT	1.00	0.99–1.01	0.558			
Post-LT 1st year	1.00	1.00–1.01	0.589			
Smoking	0.88	0.35–2.20	0.786			
Pre-LT CVD history	6.85	2.58–18.18	<0.001	6.51	2.43–17.46	<0.001
Post-LT AKI	3.07	1.41–6.72	0.005	3.06	1.39–6.75	0.006
Hypertension						
Before LT	2.31	0.92–5.83	0.075			
After LT *	0.93	0.25–4.64	0.922			
Diabetes						
Before LT	2.67	1.23–5.78	0.013	2.37	1.09–5.17	0.031
After LT *	0.00 ^&^	0.00 ^&^	0.977			
CKD						
Before LT	3.15	0.70–14.21	0.136			
After LT ^#^	3.74	1.67–8.39	0.001			
ESRD	6.08	2.09–17.67	0.001			

BMI—body mass index; CKD—chronic kidney disease; CVD—cardiovascular disease; ESRD—end-stage renal disease; HR—hazard ratio; LT—liver transplantation. * After LT means de novo hypertension or diabetes mellitus. ^#^ CKD after LT represents all LT recipients with estimated glomerular filtration rate <60 mL/min per 1.73 m^2^. ^&^ No development of MACE in recipients with post-LT DM.

**Table 4 jcm-11-03100-t004:** Risk Factors of ESRD after Living Donor Liver Transplantation by Cox Regression Analysis.

	Univariate Analysis			Multivariate Analysis		
	HR	95% CI	*p*-Value	HR	95% CI	*p*-Value
Age (year)	1.02	0.96–1.09	0.520			
Men	0.43	0.17–1.10	0.078			
Hypertension						
Before LT	2.26	0.73–7.02	0.158			
After LT *	1.57	0.35–7.04	0.533			
Diabetes						
Before LT	5.51	2.04–14.91	0.001	3.51	1.25–9.86	0.017
After LT *	1.06	0.13–8.84	0.955			
Pre-LT CKD	15.29	5.91–39.54	<0.001	9.54	3.49–26.10	<0.001
Post-LT AKI	8.01	2.63–24.43	<0.001	6.76	2.19–20.91	0.001
Post-LT AKI and pre-LT CKD						
No AKI or CKD	1.00			1.00		
AKI only	10.58	2.28–49.11	0.003	10.17	2.19–47.37	0.003
CKD only	26.70	3.76–189.68	0.001	21.18	2.94–152.69	0.002
AKI on CKD	119.52	22.75–627.93	<0.001	76.36	14.04–415.25	<0.001

* After LT means de novo hypertension or diabetes mellitus.

**Table 5 jcm-11-03100-t005:** Risk Factors of Post-LT AKI after Living Donor Liver Transplantation by Logistic Regression Analysis.

	Univariate Analysis			Multivariate Analysis		
	OR	95% CI	*p*-Value	OR	95% CI	*p*-Value
Age (year)	1.03	1.01–1.05	0.019	1.04	1.01–1.06	0.007
Men	0.95	0.63–1.43	0.808			
MELD score	1.02	1.00–1.05	0.043			
HCC	0.69	0.48–0.99	0.044	0.64	0.43–0.93	0.021
Hypertension	1.80	1.05–3.08	0.033	1.73	1.00–3.01	0.045
Diabetes	1.52	0.99–2.33	0.055			
CKD	1.93	0.87–4.25	0.104			
Hepatorenal syndrome	1.93	0.67–5.59	0.226			
Pre-LT CVD history	1.71	0.68–4.29	0.253			
Blood loss (L)	1.05	1.02–1.09	0.001	1.05	1.02–1.08	0.001
Serum albumin (g/dL)	0.68	0.50–0.92	0.011	0.72	0.54–0.99	0.041
Serum Cr (mg/dL)	1.12	0.70–1.79	0.630			
eGFR (mL/min/1.73 m^2^)	1.00	0.99–1.00	0.658			

CKD—chronic kidney disease; Cr—creatinine; MELD—model of end-stage liver disease; HCC—hepatocellular carcinoma; CVD—cardiovascular disease; eGFR—estimated glomerular filtration rate; OR—odds ratio.

## Data Availability

The data collected and analyzed during the current study was available from the corresponding author on reasonable request.

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
