# Peer review of "Excess Risk of Major Adverse Cardiovascular and Kidney Events after Acute Kidney Injury following Living Donor Liver Transplantation"

_jcm, 2022, doi:10.3390/jcm11113100_

Round 1

Reviewer 1 Report

Chan et al submitted the title, "Excess Risk of Major Adverse Cardiovascular and Kidney Events after Acute Kidney Injury Following Living Donor Liver Transplantation" exemplifies an example of precision medicine. The key highlights of the papers are demonstrating the statistical models relating AKI, MACE, MAKE of LT patients. There are some points that need to improve to increase the readability of the paper.

  1. Improve the introduction by adding a small paragraph/lines: (a) there are other factors that affect the 5 year survival of LT patients (such as genetics, secondary infections or underlying disorders etc), at least write down their names in 1-3 lines; (b) Do expand the description of the severity of MACE and MAKE in LT patients, to show readers the impact of the current study.
  2. Correct the author's affliation according to MDPI style as some information is missing.
  3. Authors used aberrations throughout the manuscript, which are hard to follow, and therefore it is advised for authors to make an abbreviation table in the end of manuscript in order to increase the reliability of the paper.
  4. In figure 3, please improve the figure by changing the color of the labeling font to black color, as to increase the visibility of the figure.
  5.  There are some typographical mistakes and formatting issues in the manuscript, do check appropriately.

The manuscript illustrates sufficient material study and methodology, therefore this manuscript would appeal to the clinical researchers working in various fields of medicine and, certainly falls in the interest of the current Journal.

Reviewer 2 Report

The manuscript is a retrospective study that shows positive correlations between post-liver transplantation (post-LT) acute kidney injury (AKI) and the incidence of major adverse cardiovascular events (MACE) and major adverse kidney events (MAKE) in patients. Additionally, pre-LT cardiovascular (e.g., diabetes) and renal (e.g., CKD) disease risk factors were predictive of MACE and MAKE, while post-AKI severity correlated with mortality. These results are consistent with the literature. The manuscript is well written, has only minor grammar errors, and the data is well presented and discussed. However, given what is known about the association of AKI with both MACE and MAKE in transplant and non-transplant population, the authors’ findings are not novel or surprising and unlikely to change current clinical practice. In addition, it should be noted that pre-LT cardiovascular and renal disease were more predictive of MACE and MAKE respectively than post-LT AKI. Other minor issues are:

Minor comments

·         Line 101-103: Please rephrase the sentence for clarity.

·         Line 120-121: Could switching some of the HCC patients from tacrolimus to rapamycin have contributed to the lower incidence of post-LT AKI in this population?

·         Line 150-151: 135 patients (75.4%, not 26.4%), 36 patients (20.1%, not 7%), 8 patients (4.5%, not 1.6%)

·         Table 3: What does a HR of 0.00 for Diabetes (After LT*) mean? Does it mean that there was no new onset diabetes in patients after LT?

·         Line 253-257: For the multivariate analysis in Table 5, do you mean odds ratio or hazard ratio?

·         Line 272: Deceased (not diseased).

·         Line 279: 35.0% (not 35.2%)?

·         Line 291: Do you mean MACE and not MAKE?

·         How many patients developed de novo diabetes post-LT?

Any comment on the role of gender in post-LT AKI incidence and MACE/MAKE?
